Taking account of uncertainties in digital land suitability assessment

Malone Brendan P. 1 brendan.malone@sydney.edu.au
Kidd Darren B. 2
Minasny Budiman 1
McBratney Alex B. 1
1 Department of Environmental Sciences, University of Sydney , Eveleigh, New South Wales , Australia
2 Department of Primary Industries, Parks, Water and Environment Tasmania , Westbury, Tasmania , Australia
Johnson Stephen
Electronic publication date: 2015 Oct 27
Publication date: 2015
Volume: 3
Electronic Location ID: e1366
Received 2015 Aug 5; Accepted 2015 Oct 9
Copyright: © 2015 Malone et al.
Copyright year: 2015
Copyright holder: Malone et al.
License: This is an open access article distributed under the terms of the Creative Commons Attribution License, which permits unrestricted use, distribution, reproduction and adaptation in any medium and for any purpose provided that it is properly attributed. For attribution, the original author(s), title, publication source (PeerJ) and either DOI or URL of the article must be cited.
License URL: https://creativecommons.org/licenses/by/4.0/

Keywords: Digital soil assessment, Digital soil mapping, Land suitability assessment, Soil mapping

Funding: ARC Linkage Project LP110200731 The research conducted for this manuscript was funded by the Australian Research Council via its Linkage Projects Scheme. ARC Linkage Project LP110200731 Wealth from Water—Soil information for new sustainable irrigated agriculture in Tasmania. The funders had no role in study design, data collection and analysis, decision to publish, or preparation of the manuscript.

==============================
Simulations are used to generate plausible realisations of soil and climatic variables for input into an enterprise land suitability assessment (LSA). Subsequently we present a case study demonstrating a LSA (for hazelnuts) which takes into account the quantified uncertainties of the biophysical model input variables. This study is carried out in the Meander Valley Irrigation District, Tasmania, Australia. It is found that when comparing to a LSA that assumes inputs to be error free, there is a significant difference in the assessment of suitability. Using an approach that assumes inputs to be error free, 56% of the study area was predicted to be suitable for hazelnuts. Using the simulation approach it is revealed that there is considerable uncertainty about the ‘error free’ assessment, where a prediction of ‘unsuitable’ was made 66% of the time (on average) at each grid cell of the study area. The cause of this difference is that digital soil mapping of both soil pH and conductivity have a high quantified uncertainty in this study area. Despite differences between the comparative methods, taking account of the prediction uncertainties provide a realistic appraisal of enterprise suitability. It is advantageous also because suitability assessments are provided as continuous variables as opposed to discrete classifications. We would recommend for other studies that consider similar FAO (Food and Agriculture Organisation of the United Nations) land evaluation framework type suitability assessments, that parameter membership functions (as opposed to discrete threshold cutoffs) together with the simulation approach are used in concert.

Introduction

It is often stated that a useful outcome from digital soil mapping (DSM) is the ability to quantify and map prediction uncertainties. Yet, as pointed out by Grunwald (2009) in a review of DSM studies, they are often not actually quantified or mapped. Or if they are, they are not really incorporated into any further analysis. It is believed they could be invaluable for a digital soil assessment (Carre et al., 2007) project. We use them in this study specifically for digital land resource or enterprise suitability assessment.

It has been observed in recent times, an increasing activity in land resource assessments that incorporate some sort of digital soil (and sometimes climate) mapping. Recent examples include Kidd et al. (2012) in Tasmania, Australia; Harms et al. (2015) in Queensland, Australia; and Van Zijl et al. (2014) in Mozambique. One reason perhaps for this activity is that one can derive with digital soil and climate modeling, very attribute specific mapping. This facilitates the opportunity for deriving quite complex suitability frameworks. These frameworks can oftentimes be specific to a particular enterprise (Kidd et al., 2012). While the suitability assessment framework design has not really progressed much more from the land evaluation guidelines prepared by the Food and Agriculture Organization of the United Nations (FAO, 1976), it is clear the suitability assessment approach is enhanced by the developments in digital soil mapping practice. An incremental advance in this area therefore is to incorporate the quantifications of uncertainty of the mapping that feed into these land or enterprise suitability assessments. It is believed that by doing this, more honest appraisals of suitability can be communicated, together with better material to use for decision making as to where efforts should be directed for improving subsequent and ongoing soil and climate mapping.

The aim of the following work is to demonstrate and evaluate one approach via a case study for incorporating input data (biophysical information) uncertainties into a land resource assessment workflow and compare with the more traditional workflow that considers the inputs to be error free.

Brief review of land suitability evaluation

For a contextual background we provide a brief overview of land and enterprise suitability evaluation. Rossiter (1996) and Mueller et al. (2010) have discussed the key concepts. Henceforth, we use the term land suitability assessment (LSA) as a collective term for its other connotations such as land evaluation, enterprise suitability evaluation or assessment or any such similar term that more-or-less conforms to the definition given by Van Diepen et al. (1991) as “all methods to explain or predict the use potential of land”. While there is a blurry concept of what ‘land’ is or means, we take its meaning from Triantafilis & McBratney (1993), paraphrasing from Brinkman & Smyth (1973) as “all reasonably stable, or predictably cyclic attributes of the biosphere above and below the earth surface, including those of the atmosphere, the soil and underlying geology, the hydrology, floral and faunal populations, and the result of previous human activity in which the effects significantly influence present or future land uses by man (sic)”.

Soil generally features heavily in a LSA with the reason possibly traceable to the rudimentary beginnings of soil science (and pedology) and subsequent grouping and classification of soils on the basis of their productivity (Brevik & Hartemink, 2010). Incorporation of climate information into LSA is also apparent. Climatic information can include variables such as temperature and precipitation (Zabel, Putzenlechner & Mauser, 2014), to other more complexly derived variables such as frequency of frost occurrence (e.g., Kidd et al., 2012). Less common is the incorporation of socioeconomic factors (D’Haeze et al., 2005). In general, greater understanding is needed in the area of coupled human and biophysical systems (Stuart et al., 2015). Intuitively, socioeconomic factors would feature heavily in a LSA because they compel activity despite biophysical constraints. Some factors include commodity price changes, institutional reforms and reforming of trade practices and agreements amongst governments and organisations. However, it is not altogether clear how these types of factors can be incorporated in the more common biophysical centric LSAs as they are often very localized to a particular area, and difficult to track.

LSA projects are practiced throughout the world due principally to the need for better management of natural resources which are finite—yet the pressures and demands emplaced upon them are increasing—for example, through human population increase. Underlying this is a fundamental need to determine whether a given site or locality would be productive or unproductive for a given land use, and then secondly evaluate its biophysical status to determine what the limitations are, and what the land may be best suited for. Broad global LSA programs include agro-ecological zoning (Fischer & Sun, 2001) and the Fertility Capability Classification (Buol et al., 1975). More recently a global high resolution LSA for a number of arable cropping enterprises together with future projections was presented by Zabel, Putzenlechner & Mauser (2014).

Mueller et al. (2010) describe numerous examples of LSA programs that are nationally rolled out throughout the world. There are also LSA projects that are quite locally situated. For example, LSA projects have been developed in both Northern Australia (Queensland) and Tasmania respectively as described in Harms et al. (2015) and Kidd et al. (2012) for enhancing agricultural pursuits in those areas.

While LSA may have had its origins in the early development of soil science and agricultural land capability classifications, the terminology and general framework were formalized with the land evaluation guidelines prepared by the Food and Agriculture Organization of the United Nations (FAO) in 1976 (FAO, 1976). These guidelines have strongly influenced and continue to guide LSA projects throughout the world. They have also informed the development of automated LSAs via such software as ALES (Automated land Evaluation System, Rossiter, 1990) and MicroLEIS (De la Rosa et al., 2009) as examples. At its core, the FAO framework is a crop specific LSA system with a 5-class ranking of suitability (FAO Land Suitability Classes) from 1: Highly Suitable to 5: Permanently Not Suitable. Given a suite of biophysical information from a site, each attribute is evaluated against some expert-defined thresholds for each suitability class. The final evaluation of suitability for the site is the one in which is most limiting. There are a number of variants to this approach such as the number of suitability classes, the weighting attributed to the inputs, and also the degree of complexity of the crop specific suitability criteria. This last variant is usually determined by availability of data, which is also related to spatial scale at which the LSA is applied. As we have seen recently however in Zabel, Putzenlechner & Mauser (2014) and with high resolution global soil mapping efforts such as the GlobalSoilMap project (Arrouays et al., 2014), the scale issue will not be as important into the future. A further generalisation of the FAO framework is evidenced by the incorporation of fuzzy threshold values for suitability parameters, as opposed to discrete threshold values (Lark & Bolam, 1997; Triantafilis & McBratney, 1993; Zabel, Putzenlechner & Mauser, 2014). Fuzzy cut-offs are intuitively sensible because one cannot easily justify (as an example) that a crop will or will not grow if a subsoil has pH of 5.4 as opposed to 5.5, given the complexity of soil and the interactions of other soil physical and chemical parameters in the system. These ideas are explored in Lark & Bolam (1997) with examples.

Other LSA frameworks include parametric grading systems that may be multiplicative or additive which ultimately rate the suitability of a land on a continuous scale rather than as discrete classes. The Storie index rating (Storie, 1932) of soils is an example of a multiplicative parametric LSA. A more data driven approach to the Storie Index for digital soil applications is described in O’Geen, Southard & Southard (2008). An additive parametric system was developed for NSW, Australia by Zhang (1989). The last LSA frameworks to be discussed are those associated with crop growth and ecosystem models that estimate local productivity of specific crops given soils and weather information (Mueller et al., 2010). These models are more sensitive to temporal variation, which make them a more dynamic LSA approach to those discussed already. While this may seem advantageous, their dynamic behavior can only be realized given appropriate input information which is often difficult to obtain. Mueller et al. (2010) describes a number of candidate models of this type that are applicable at different scales from global to local extents. The widespread application of these models in LSA projects is limited due to their sophistication and inflexibility about input data requirements as many are designed for a specific purpose and applicable only to their origin.

Other than fuzzy classification methods (Triantafilis & McBratney, 1993; Zabel, Putzenlechner & Mauser, 2014), the assumption of LSAs is that the input data and parameter thresholds are free of error. However, soil is complex and can vary quite erratically in the context of space and time (Webster, 2000), and subsequent model-based predictions of soil phenomena are anything but ‘error free’ (Brown & Heuvelink, 2005). In noting this, Harms et al. (2015) provided additional mapping of a confidence measure (based on Mahalanobis distance calculations) of the suitability classifications. They were able to state explicitly where suitability classifications were likely to be good and where they were likely to be uncertain. Indeed this is helpful and a good step forwards, but possibly a more direct way to derive a measure of land suitability is to incorporate the measures of prediction uncertainty into the final assessment.

Materials and Methods

The case study is situated in the Meander Valley, in north-east Tasmania (Fig. 1). This area, together with the Midlands irrigation district in central Tasmania is a focus site of the ‘Wealth from Water’ (WfW) project that commenced in November 2010 (Kidd et al., 2012). The Meander study area has diverse soils and landscapes; the eastern extremities are part of the Launceston tertiary Basin (Doyle, 1993), with mainly duplex profiles (sharp change in texture between the A and B horizons, and sodic subsoils (exchangeable sodium % > 6), Sodosols (Australian Soil Clasification (ASC) Isbell, 2002)); Lixisols or Solonetz (World Reference Base (WRB) IUSS Working Group WRB, 2007)). Productive areas of deep, gradational Tertiary Basalt soils are formed around Deloraine (Ferrosols (ASC); Nitisols or Acrisols (WRB)), and highly complex, poorly drained alluvial soils to the south near Meander (Hydrosols, Kandosols, Chromosols (ASC); Gleysols, Fluvisols and Lixisols (WRB) (Spanswick & Zund, 1999).

Figure 1 Study area map.

Locality map of the Meander Valley study area, Tasmania.

Current land use is mainly grazing, cereal and vegetable cropping in the east, and dairying in the west and south, with forestry and conservation in rocky and mountainous areas. Average annual rainfall is approximately >800 mm/yr.

The aim of the WfW project was to develop a LSA for a number of agricultural enterprises to support irrigated agricultural expansion across the state (Kidd et al., 2012). Kidd et al. (2014a) and Kidd et al. (2014b) describe the varied functions of this project which have included extensive soil sampling and climate monitoring, together with digital soil and climate modeling to support an LSA for 20 listed agricultural enterprises.

This case study describes an LSA for hazelnuts in the Meander Valley with two contrasting approaches. The first approach (Approach 1) is an LSA that considers the input variables to be error free. While the second approach (Approach 2) is an LSA which takes into consideration the prediction uncertainties of the input variables. The commonality between both approaches is the underlying assessment—it being based on the FAO most limiting factor model of land evaluation. As with all agricultural enterprises investigated for the WfW project, hazelnut suitability thresholds were determined by experts from the Tasmanian Government Department of Primary Industries, Parks, Water and Environment (DPIPWE), together with input from the Tasmanian Institute of Agriculture as well as from research trial information, existing literature and consultation with industry experts. The suitability thresholds are related to biophysical information pertaining predominantly to soil and climatic information, and are tabulated in Table 1. Soil and climate variables were predicted using digital soil mapping methods (McBratney, Mendonca Santos & Minasny, 2003). Details of the methods used for preparing and modeling the soil variable data are described in Kidd et al. (2012), while the approaches used for the climate variables are described in Webb et al. (2014) and Webb et al. (2015). A short description of the modeling approaches used for each soil and climate variable follows in the next section, but the main details are summarized in Table 2, together with the model validation diagnostics of both the predictions and quantifications of uncertainty.

Table 1 Hazelnut suitability parameters.

Suitability parameters and thresholds for hazelnuts. Sourced from DPIPWE (2015).

Soil variables	Climate variables	
Suitability class	Soil depth (cm)	Soil pH (1:5 soil water) top 15 cm of soil	Soil conductivity (dS/m) top 15 cm of soil	Soil texture (% clay) top 15 cm of soil	Soil drainage	Stoniness (>2mm) of the top 15 cm of soil	Incidence of frost (Probability of zero days in June, July or August where temperature reaches −6 °C)	Mean maximum monthly temperature. Mean January or February max temp (°C)	Rainfall (March mean)	Chill hour requirements (chill hours 0–7 °C April–August inclusive)	
Well suited	>50 cm	5.5–6.5	<0.15	10–30	Well to moderately well drained	<10%	≥ 80%	20–30	<50 mm	>1,200	
Suited	40–50 cm			30–50	Imperfectly drained	10–20%	60–80%	30–33 and 18–20		600–1,200	
Marginally Suited	30–40 cm	6.5-7.1					40–60%	33–35			
Unsuited	<30 cm	<5.5 or >7.1	>0.15	>50 or <10	Poor to very poorly drained	>20%	<40%	>35 or <18	>50 mm	<600	

Table 2 Model diagnostics.

Metadata and model diagnostics of digital soil and climate mapping relevant to LSA for hazelnuts.

Variable	Number of observations	Model used	Residual modeling (variograms)	Validation statistics	PICP	
Soil depth (cm)	1. 432 (144) 2. 56 (20)	1. Binomial logistic regression 2. Cubist model regression kriging (1 rule)	2. Yes	1. OA = 87%, Kappa = 0 2. RMSE = 16.2, CCC = 0.31	2. 92%	
Soil pH (1:5 soil water) top 15 cm of soil	432 (144)	Cubist model regression kriging (1 rule)	Yes	RMSE = 0.44, CCC = 0.25	92%	
Soil conductivity (dS/m) top 15 cm of soil	426 (143)	Cubist model regression kriging (1 rule)	Yes	RMSE = 0.12, CCC = 0.09	81%	
Soil texture (% clay) top 15 cm of soil	269 (94)	Cubist model regression kriging (1 rule)	Yes	RMSE = 7.77, CCC = 0.38	90%	
Soil drainage	431 (144)	Cubist model regression kriging (1 rule)	Yes	RMSE = 0.70, CCC = 0.52	91%	
Stoniness class (>200 mm) of the top 15 cm of soil	1. 432 (144) 2. 46 (18)	1. Binomial logistic regression 2. Ordinal logistic regression		1. OA = 85%, Kappa = 0.15 2. OA = 34, Kappa = 0		
Incidence of frost (Probability of zero days in June, July or August where temperature reaches −6 °C)	1. 129 (44) 2. 14 (5)	1. Binomial logistic regression 2. Multiple linear regression model	2. No	1. OA = 91%, Kappa = 0.30 2. RMSE = 25, CCC = 0	2. 60%	
Mean maximum monthly temperature. Mean January or February max temp (°C)	129 (44)	Cubist regression model (1 rule)	No	RMSE = 0.60, CCC = 0.62	88%	
Rainfall (March mean)	21	Multiple linear regression	No	RMSE = 3.23, CCC = 0.69	90%	
Chill hour requirements (chill hours 0–7 °C April-August inclusive)	129 (44)	Cubist regression model (2 rules)	Yes	RMSE = 54, CCC = 0.84	82%	

Digital soil and climate modeling for land suitability assessment

An extensive soil sampling and climate modeling program was established in the Meander Valley in 2010 (Kidd et al., 2012; Kidd et al., 2015). In total 576 soil cores were extracted from various locations throughout the area and analysed for a number of physical and chemical soil properties both in the laboratory and with chemometric techniques dependent on mid-infrared soil spectral calibrations (Kidd et al., 2015). Climate and temperature monitoring throughout the Meander was carried out over 2010 and 2011 using a network of distributed temperature sensors (Webb et al., 2014), and calibrated to long-term climate data (Webb et al., 2015). Rainfall information was sourced from Australian Bureau of Meteorology monitoring sites located within and surrounding the study area (Webb et al., 2014). Soil sampling and climate monitoring network were conducted and installed respectively by field officers of the Tasmanian Government. For many sites that were situated on public or Government administered land, no permission was required for sampling or installation of temperature sensors. For sites that were situated on private landholdings, Government officers sought permission for access. In the rare situation where permission was not granted, the sampling or monitoring site was moved to an alternative and agreeable location.

The common workflow for all digital soil and climate variable mapping entailed:

1. The randomized splitting of observational data into calibration and validation datasets. Here a 75% and 25% split was used respectively for calibration and validation datasets. For consistency, the same calibration and validation datasets (soil, climate, rainfall) were used for all target variables, before removal of missing values.

2. Environmental covariates were sourced from DPIPWE and other government repositories which included derivatives from a digital elevation model (STRM DEM (Gallant et al., 2011)), gamma radiometric information (Minty et al., 2009), and spectral indices derived from Landsat 7 ETM+ satellite. Soil modeling involved using principal components of all sourced covariates, while principal components of the digital elevation model derivatives (only) were used for the climate variables (Webb et al., 2015).

3. Modeling of continuous variables was based on a regression kriging framework that entailed Cubist regression tree modeling (Quinlan, 1992) followed by model residual modeling (with variograms) and kriging. Spherical or exponential models were considered only. Visual criteria of the global variogram of residuals were used to determine whether regression kriging should be pursued or not. Otherwise regression modeling was used only. Categorical variable modeling entailed either the fitting of binomial or ordinal logistic models, dependent on the nature of the target variable information.

4. Prediction uncertainties for continuous variables were quantified using an empirical approach as described in Malone et al. (2014) where the model errors within each partition of a Cubist model were used to form geographically specific error distributions (via leave-one-out cross validation) in order to estimate 90% prediction intervals. For categorical variables, the prediction probabilities were used as measures of uncertainty.

5. Validation statistics for continuous variables included the root mean square error (RMSE) and Lin’s concordance correlation coefficient (CCC; Lin, 1989). The prediction interval coverage probability (PICP, Shrestha & Solomatine, 2006) was used to evaluate the efficacy of the 90% prediction intervals. The PICP is the proportion of actual observations that are encapsulated by their prediction interval, and ideally will be equivalent the level of confidence associated with the prediction interval. For categorical variables, overall accuracy and kappa statistic (Congalton, 1991) were used. Validation statistics are reported for both calibration and validation datasets. The PICP is reported for the validation data set only.

All maps of the biophysical properties (together with uncertainties) used for the hazelnut LSA are supplied as Supplemental Information 1 to this research. Some details specific to each modeling variable are as follows.

Soil depth

Soil depth was modeled in a two-step procedure. The first step was binomial modeling of whether soil depth greater than 1.5 m or not. The rationale behind this was that soil coring was done to a maximum of 1.5 m depth, or depth-to-lithic contact, whichever was first. The second step involved regression kriging modeling of soil depth where soil depth was less than 1.5 m. The outputs from both steps were used for the LSA, which is discussed further on.

The two-step modeling procedure described in this study has been used previously by Gastaldi, Minasny & McBratney (2012) for mapping the occurrence and thickness of soil horizons. In that study, the first step involved modeling the occurrence of horizon classes, while the second step involved modeling the depth of the soil horizons.

Soil pH, EC and clay

A mass-preserving soil depth spline (Bishop, McBratney & Laslett, 1999) was used to harmonise observed soil profile data in order to impute data for the 0–15 cm depth intervals for all locations. Spatial modeling was performed using regression kriging framework.

Soil drainage

Digital soil mapping of soil drainage was carried out as in the method described by Kidd et al. (2014a) and Kidd et al. (2014b). Essentially this method codifies the descriptive soil drainage classification as detailed in The National Committee on Soil and Terrain (2009) into an ordinal value. The numerical classes were then spatially modeled as a continuous variable using regression kriging.

Stoniness

Soil stoniness or percentage of coarse fragments greater than 2 mm was modeled in a two-step procedure, much for the same reasons as for soil depth. Here the first step was a binomial logistic regression model which considered the presence vs. absence of coarse fragments. The second step entailed an ordinal logistic regression of the coarse fragment incremental percentage classes (6) as described in The National Committee on Soil and Terrain (2009).

Incidence of frost

The method for estimating the probability of frost incidence at each location is described in Webb et al. (2014) and Webb et al. (2015). Spatial modeling occurred as a two-step procedure because many sites had no incidence of frost occurrence. Subsequently the first step entailed a binomial logistic regression of presence vs. absence of frost occurrence followed by the second step of regression kriging of frost occurrence probability using the sites where information relating to this was recorded.

Temperature, rainfall and chill hour requirements

These were all spatially modeled as continuous variables. A multiple linear regression model was used for rainfall modeling, due to the small number of rainfall observations. The validation statistics reported are from a leave-one-out-cross validation. The PICP for the validation of prediction uncertainties is based on the calibration data. Cubist regression models were used for temperature and chill hours, with residual kriging being appropriate for chill hours only.

Approach 1. Land suitability assessment without considering prediction uncertainties

Raster maps of each of the hazelnut LSA variables were interrogated pixel-by-pixel from which an assessment of suitability was derived using the most limiting factor approach. For variables that were modeled as a two-step procedure (soil depth, stoniness, frost incidence), a positive condition invoked the interrogation of the map from second part of the two-step procedure. Using the soil depth variable as an example, if a pixel on the prediction map recorded a positive score (likely presence that depth to lithic contact is less than 1.5 m) contact, the second map was interrogated to estimate the depth to the lithic contact.

Approach 2. Land suitability assessment in consideration of prediction uncertainties

The basis of the approach is a simulation of possible realisations (pixel-by-pixel) of the input variables before assessing the suitability using the most limiting factor approach. For the continuous variables the prediction intervals were sampled with an assumption of normal distribution upon each realisation. For the categorical variables, sampling from either the binomial or multinomial distributions of the prediction probabilities was performed. In consideration of variables subject to the two-step model procedure, the second condition was invoked where a presence or occurrence of the phenomenon in question was found for a given sample. If the condition was invoked, the probability distribution of the second variable was sampled.

There were two issues to consider in this approach. The first was an issue of computation where compared to approach 1, the computation time can increase dramatically with increasing number of realisations. Besides using multi-core compute facilities, Latin Hypercube Sampling (LHS) was used to sample from the multivariate distributions of the continuous variables. LHS sampling is a more efficient sampling approach to random sampling if the objective is to ensure the multivariate distributions are sampled entirely (Pebesma & Heuvelink, 1999).

The second issue was one of rationality in terms of the multivariate information generated from each realisation. The rationality here was the maintenance of correlations between the LSA variables. Subsequently a modified LHS was used where the correlation matrix of the sample is corrected to match that of the data that is being sampled. The correction was made through the use of the Huntington–Lyrintzis algorithm (Huntington & Lyrintzis, 1998). Implementation of correlation matrix corrected LHS was performed using of the LHScorcorr function in the pse R package (Chalom & Prado, 2014).

In this study 100 realisations were made in order to estimate probabilities of each suitability class at each map pixel. The probabilities were estimated as the number of occurrences of each suitability class divided by the total number of realisations. Further analysis of these outputs is presented in the results.

Software

The entire workflow of this study was implemented in the R scripting language (R Core Team, 2015) except for the creation of map products which were created using ArcGIS (ESRI Inc.). Specifically, custom functions were built for the hazelnut LSA. The raster (Hijmans, 2015) package with associated rgdal (Bivand, Keitt & Rowlingson, 2015) and sp (Bivand, Pebesma & Gomez-Rubio, 2013) packages were used for handling and manipulating all GIS processes. Cubist models were implemented from the Cubist (Kuhn et al., 2014) package and the multinomial and ordinal logistic modeling were carried out using the nnet and MASS packages respectively (Venables & Ripley, 2002). Variogram modeling entailed the usage of geostatistical functions from gstat (Pebesma, 2004). Many of the processes, including applying models spatially and applying the LSA in the presence of prediction uncertainties were distributed across multiple compute nodes (8) to improve the computation efficiency. This was done using specialist R packages parallel (R Core Team, 2015) and doParallel (Revolution Analytics & Weston, 2014).

Results

The suitability map for hazelnuts calculated without considering uncertainty is shown in Fig. 2. This map indicates that approximately 6% of the study area has the biophysical characteristics that are well suited for hazelnuts. 56% was classified as suitable, while 20% and 18% respectively were classified as marginally suited and unsuited. The classification of ‘unsuited’ within the area was predominantly on the basis of soil pH. Those areas where unsuited is classified have a predicted low topsoil pH (<5.5). The areas where marginal suitability was classified occur where the incidence of frost is reasonably common i.e., there is between 40 and 60% likelihood that there will be no days in June, July or August (Southern Hemisphere Winter) where temperature reaches −6 °C. From this analysis, there were very few situations where there were multi-factor issues causing limitations. One exception however was in the northern peninsula area where both soil pH and frost limitations were predicted. Maps showing the limitation classifications for each soil and climate LSA parameter are to be found in the Supplemental Information 2 of this research manuscript.

Figure 2 Error free suitability classification.

Hazelnut suitability classification assuming LSA inputs are error free.

The main outputs from approach 2 are the four maps shown in Fig. 3. This shows the probability maps for each suitability class: (3A) well suited, (3B) suited, (3C) marginally suited and (3D) unsuited. The mean probability for each suitability class (in the same sequential order as before) is 6%, 21%, 7%, and 66% respectively. From this result, taking into consideration the prediction uncertainties of the LSA parameters changes the overall assessment that was made using approach 1. The principle reason for a high likelihood of an unsuitable classification from approach 2 is because of the magnitude of the input data uncertainties (which are not taken into account in approach 1), and how they are propagated through the LSA.

Figure 3 Hazelnut suitability class probabilities.

Hazelnut suitability class probabilities (A) Well suited, (B) suited, (C) marginally suited, (D) unsuited.

Figure 4 shows maps of the probabilities to which each LSA parameter contributes to an ‘unsuitable’ assessment for hazelnut suitability. For illustrative purposes, a probability threshold of 0.6 was used to delineate from Fig. 3D, areas where it is reasonably certain that hazelnut suitability is low. It is clear from Fig. 4 that the main contributors are soil pH (4F) and soil conductivity (4D). Soil texture (4B) and rainfall (4G) impart a minor contribution to the unfavorable suitability estimate. Figure 5 shows the comparison between approach 1 and approach 2 in terms of the contribution of soil conductivity (5A and 5C) and soil pH (5B and 5D) to the suitability estimates. Figures 5C and 5D are the same as in Figs. 4D and 4F respectively. From approach 1, soil conductivity imparts no limitation, but from approach 2 it clearly is a main contributor. The reason for this is simple: the spatial prediction of soil conductivity is currently highly uncertain. The RMSE of the soil conductivity map was 0.12 dS/m, while the threshold to delineate between well suited and unsuited was 0.15 dS/m. Subsequently, a large number of realisations using approach 2 would sample values above this threshold. This is similarly the case for soil pH, but to a lesser degree. Here the RMSE of the soil pH map was 0.44, and coupled with relatively narrow threshold criteria for suitability classification, there is going to be a relatively high likelihood that it will contribute unfavorably to suitability estimates. Despite the magnitude of uncertainty, areas where soil pH is a limitation on Fig. 5B are also observable from Fig. 5D. The difference is that the spatial extent of soil pH limitation in Fig. 5B is less constrained than in Fig. 5D.

Figure 4 Probabilities of each soil and climate variable causing hazelnuts to be unsuitable for establishment.

Probabilities of each soil and climate variable causing hazelnuts to be unsuitable for establishment, (A) chill hours, (B) clay content, (C) soil drainage, (D) soil conductivity, (E) frost occurrence, (F) soil pH, (G) rainfall, (H) soil depth, (I) temperature, (J) rock fragments.

Figure 5 Contribution of soil conductivity and pH to assessment of unsuitability for hazelnuts.

Contribution of soil conductivity and pH to assessment of unsuitability for hazelnuts (A) and (B) assuming inputs are error free, (C) and (D) taking account of the associated prediction uncertainties.

As with approach 1, there was not a widespread incidence of multifactor issues from approach 2. Figure 6 illustrates this which shows the probability of situations where more than one LSA parameter contributes to an unfavorable suitability prediction. Mostly where the probability is high, it is soil pH and conductivity that are the main contributing factors. Some areas in the middle of the study area (to the southern edge), soil texture is also a contributing factor, together with pH and conductivity.

Figure 6 Probability of multiple factors.

Probability of multiple factors that contribute towards unsuitability for hazelnut establishment.

General Discussion

This research has been a case study to explore an enterprise LSA given uncertain input variables. Taking into account the biophysical variable prediction uncertainties is a slight sophistication to many LSA analyses which mainly consider inputs to be error free. One caveat to this is that the workflow becomes more computationally demanding—due to the requirement to run multiple realisations—and creates a significant number of items which need to be managed accordingly. For example the spatial modeling and uncertainty quantification of LSA inputs requires a significant amount of effort and organisation. Some of this workflow can be made more efficient however through parallel and high performance computing abilities which are becoming more prolific in scientific studies today.

It has been demonstrated (using the LSA for hazelnuts) that the prediction uncertainties of inputs can significantly alter the LSA outlook compared to the situation where they are not considered (approach 1). In this study, optimistic results from approach 1 were counter-matched with less favorable outcomes for approach 2, despite the same information being used. As stated in Harms et al. (2015), by explicitly quantifying the uncertainty of the LSA inputs, an ability to assess the quality of suitability assessments can be realized. This is important for strategic decision making regarding land resource management. Consequently this research highlights the importance of soils (Bouma & McBratney, 2013)—in this situation for a LSA—yet it has been made apparent that more needs to be done to improve the predictive grasp of soil spatial predictive functions that will act as inputs for associated biophysical models. It has been considered before that some ways of doing this include additional soil sampling and discovery of more informative predictive covariates (Malone et al., 2014). This may well be true for most cases, and as Thomas et al. (2015) point out, digital soil mapping is a dynamic exercise, where a prediction model (and map) is never final. With new information, new calibrations and updates can be made (Kempen, Brus & De Vries, 2015) with the long view that upon each iteration, one gets a better and more sure grasp of the phenomenon being modeled.

The LSA for hazelnuts in this study using approach 2 serve a purpose and use despite the known issue of some uncertain inputs. It was established that in consideration of the other LSA variables besides soil pH and conductivity, there do not appear to be any major limitations for hazelnuts. A pragmatic solution for a landholder to gain utility from the suitability mapping would be to conduct further testing of these soil attributes if they are believed to be an issue. Relative to other soil attributes the cost of measuring soil pH and conductivity is minimal.

Despite the need to make improvements in both digital soil and climate modeling, this study has perhaps illustrated the limitations of using discrete thresholds for LSA parameter suitability criteria. With the example used in this study, the multiple realisations do not consider or adjust for those values that are near and just slightly breach the criteria that would give an unfavorable suitability outcome. The severity of the limitation in terms of management potential needs considering. For example, soil pH, if close to a threshold, can easily be managed by the application of lime. Whereas a limitation such as frost would be harder to manage. An approach such as parametric weighting could provide a more realistic suitability framework. Furthermore, while there is no disputing the expertise of the practitioners who develop biophysical parameter thresholds, another (intuitive) approach is to develop membership functions rather than discretized functions of the parameters thresholds when designing an enterprise LSA (Zhang et al., 2015). This would effectively incorporate the uncertainty in defining what the threshold values might be. This approach will in turn, circumvent those situations of borderline classification that clearly can have misleading outcomes on the overall suitability assessment. As was previously established, membership functions for LSA are not unheard of, with the recent study by Zabel, Putzenlechner & Mauser (2014) being one such example. Ultimately, due consideration of this additional source of uncertainty (threshold uncertainty) would make a modest improvement upon the continuous suitability assessments made in this study, despite the uncertainties of the LSA inputs.

Conclusions

• Taking account of the uncertainties adds to the overall LSA because one can actually assess the reliability of the assessment.

• Because the input variables are generated through a digital soil mapping approach, there is an ability to continually update the mapping as a means to improve accuracy, which will in turn, yield a more reliable LSA.

• Consideration of the biophysical variable uncertainties can have a significantly different LSA outcome to when they are not.

• With the approach proposed in this study, it is possible to identify and assess the magnitude to which biophysical variables contribute most to a classification of ‘unsuitable’ in a LSA.

• Truly incorporating uncertainties into an LSA would also include the incorporation of membership functions rather than discrete thresholds for each of the biophysical input variables.

• While there are many variants of a LSA, they are fundamentally quite similar. Therefore we would suggest they could all be adapted for simulation studies as shown in this study in order to derive continuous rather than discrete assessments of land suitability.

Supplemental Information

Supplemental Information 1 All DSM predicted maps of the biophysical properties (together with uncertainties) used for hazelnut LSA.

Click here for additional data file.

Supplemental Information 2 Maps showing the limitation classifications for each soil and climate LSA parameter (assuming inputs to be error free).

Click here for additional data file.

Additional Information and Declarations

Competing Interests

Author Contributions

Data Availability

Budiman Minasny is an Academic Editor for PeerJ.

Brendan P. Malone conceived and designed the experiments, performed the experiments, analyzed the data, wrote the paper, prepared figures and/or tables, reviewed drafts of the paper.

Darren B. Kidd conceived and designed the experiments, performed the experiments, analyzed the data, contributed reagents/materials/analysis tools, reviewed drafts of the paper.

Budiman Minasny and Alex B. McBratney conceived and designed the experiments, reviewed drafts of the paper.

The following information was supplied regarding data availability:

University of Sydney e-Scholarship Repository: http://hdl.handle.net/2123/13815.

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
