# Peer review of "Taking account of uncertainties in digital land suitability assessment"

_PeerJ, doi:10.7717/peerj.1366_

## Round 0.1 · original submission · Major Revisions

The results are presented well, and the overall organization of the manuscript is good, but the methods section needs to be made much clearer.

See the reviewers' comments about how uncertainty was obtained, in particular in the soil depth. Also, it is unclear from the text how the dataset was split between training and validation data. Did you do this randomly in R? If so, did you use set.seed() to ensure reproducibility? Did you consider other approaches such as leave-p-out or leave-one-out?

Please ensure that the original datasets used to generate the results are made available, either alongside the manuscript, or in a publicly-available repository. Please also provide the R scripts, along with the version of R and any packages used to allow reproduction of the figures - this information can be found by invoking sessionInfo().

There are some awkwardly-phrased sections that should be rewritten so that their meaning is unambiguous. In particular, the abstract, and Lines 91-93, 129, 145 (There -> Their), 184-185, 217 (sort -> sought), 263, 381-382. Please note also reviewer comments regarding the use of overly subjective language, and be sure to spell out abbreviations on first use (FAO, RMSE, DPIPWE) .

Reviewer 1 ·

Basic reporting

This paper attempts to quantify the uncertainty in landscape suitability assessments for hazelnuts in Tasmania. This is a good goal and the results are decently well-presented. However, the Methods are not clear and overall the paper is poorly written.

There are many subjective statements (e.g. “ringing declarations” and “without doubt”) that aren’t well-suited to a scientific publication. Many statements are unclear, such as this one from the abstract:

“Using the simulation approach there is approximately 50% likelihood across the whole area that hazelnut establishment would be unsuitable.”

Does this say that the sum of probabilities for every pixel does not exceed 0.5? Or that no pixel had a value greater than 0.5? Or that less than 50% of the total area was deemed suitable? Based on reading the manuscript, I think the authors mean the last.

Experimental design

Besides the distractingly poor writing, it is not clear how uncertainty was generated. The author describes preparation of input characteristics for the LSA, but they are, in my opinion, unintelligible. For example, here is the one for soil depth:

“Soil depth was modeled in a two-step procedure. The first step was binomial modeling of whether soil depth greater than 1.5m or not. The rationale behind this was that soil depth is likened to depth to lithic contact and as a consequence of sampling to 1.5m, in many cases lithic contact was not observed. Here the assumption is where soil depth was not recorded; the soil is in fact greater than 1.5m. The second step involved regression kriging modeling of soil depth where soil depth was less than 1.5m. The outputs from both steps were used for the LSA, which is discussed further on.”

What does “likened to” mean? Is lithic contact the independent variable? “In some cases, lithic contact was not observed”, so then what happened? A kriging? Supposing for the moment that I should be able to understand the methodology based on this passage, I still have no idea of how this method leads to uncertainty in this input parameter. In other words, based on this methodology, how different are potential soil depth maps?

Validity of the findings

Without understanding how big the uncertainty is in soil depth (preferably with some sort of graphic), I am not sure how I am supposed to determine if the results are reasonable (as in Fig. 4h where soil depth does not contribute much to the classification as “unsuitable”).

Reviewer 2 ·

Basic reporting

The manuscript is well organized
The work is a good approach towards the wise and sustainable use of land
But the discussion section need little more attention
Introduction and Method section is well written

Experimental design

Experimental design is good

Validity of the findings

No comments

Additional comments

Just add a key line to the abstract.
Needs little reconstruction of sentences, eg., line no. 91-92
Please consider cases with similar approach but with different problem and compare their results with yours.
Please reconsider the punctuation marks
Make the conclusion more precise and some portion of it could be integrated with discussion section.

---

## Round 0.2 · Minor Revisions

There remain some minor issues of English that should be addressed before publication. Please do a final run through for grammar and punctuation - I would suggest the following but you should also check for anything I may have missed:

L28 This study was...
L29 It was found that compared to a...
L32 ...it was revealed...
L147 The widespread application of these models in LSA projects...
L151 However, soil is...
L154 Delete comma after "but"
L156 Sentence beginning "They were able..." mixes tenses (were/are)
L164 Consider adding a reference for the WfW project
L173 Check capitalizations of Land, East, West, South
L185 Confusing sentences ("considering and compares")
L186 Rewrite sentence beginning "Approach 1..."
L194 "Both the soil and..." -> "Soil and..."
L204 Consider moving the URL to the reference section and adding a citation here.
L208 "About 576 soil cores" seems rather specific.
L216 Soil sampling and climate monitoring network were...
L267 Insert space between a quantity and its unit (e.g. 1.5m -> 1.5 m). See also e.g. L267, 268, 277, 286, 397, 398 and Column 1 of Table 2.
L271 Consider rewriting so that model descriptions agree (e.g. step 1/step 2 or first step/second step)
L283 them -> then
L303 leave-one-out cross validation
L310 which -> that
L325 "approach 1," -> "approach 1;" (or start a new sentence)
L333 The correction was made...
L357 The suitability map for hazelnuts calculated without considering uncertainty....
L363 It might be worth reminding readers that June-August are winter months in the southern hemisphere (just a suggestion - the majority of readers are in the north)
L387 The principle reason for a high likelihood of an unsuitable classification from approach 2 is the magnitude of...
L417 As with approach 1...
L420 For some areas in the middle...
L431 One caveat...
L448 ...some ways of doing this include additional...
L450 Thomas et al. ...point out... (Thomas et al. are plural)
L454 It was established that....
L458 "if these are" -> "if they are"
L475 Insert comma after "Ultimately"
L481 The first bullet point is not a conclusion
L488 Delete "It is clear from this study" and capitalize "Consideration"
L514 Check spacing of references
L643 Delete comma after authors. Place parentheses around year.
L688 Plos -> PLOS

---

## Round 0.3 · accepted · Accept

Thank you for attending to the minor revisions to the previous version. I hope you found the review process to be constructive.